# Nanotechnology as a Promising Method in the Treatment of Skin Cancer

**DOI:** 10.3390/ijms25042165

**Published:** 2024-02-10

**Authors:** Angelika A. Adamus-Grabicka, Pawel Hikisz, Joanna Sikora

**Affiliations:** 1Department of Bioinorganic Chemistry, Faculty of Pharmacy, Medical University of Lodz, Muszynskiego 1, 90-151 Lodz, Poland; angelika.adamus@umed.lodz.pl; 2Department of Oncobiology and Epigenetics, Faculty of Biology and Environmental Protection, University of Lodz, Pomorska 141/143, 90-236 Lodz, Poland; pawel.hikisz@biol.uni.lodz.pl

**Keywords:** skin cancers, drug delivery, liposomes, polymeric nanocarriers, nanoparticles, nanofibers, carbon nanotubes

## Abstract

The incidence of skin cancer continues to grow. There are an estimated 1.5 million new cases each year, of which nearly 350,000 are melanoma, which is often fatal. Treatment is challenging and often ineffective, with conventional chemotherapy playing a limited role in this context. These disadvantages can be overcome by the use of nanoparticles and may allow for the early detection and monitoring of neoplastic changes and determining the effectiveness of treatment. This article briefly reviews the present understanding of the characteristics of skin cancers, their epidemiology, and risk factors. It also outlines the possibilities of using nanotechnology, especially nanoparticles, for the transport of medicinal substances. Research over the previous decade on carriers of active substances indicates that drugs can be delivered more accurately to the tumor site, resulting in higher therapeutic efficacy. The article describes the application of liposomes, carbon nanotubes, metal nanoparticles, and polymer nanoparticles in existing therapies. It discusses the challenges encountered in nanoparticle therapy and the possibilities of improving their performance. Undoubtedly, the use of nanoparticles is a promising method that can help in the fight against skin cancer.

## 1. Introduction

At the end of 2020, the International Agency for Research on Cancer published alarming data from the global cancer burden (GLOBOCAN), which showed that in 2020 the number of newly diagnosed cancer cases increased to 19.3 million, and the total number of deaths during this period was almost 10 million [1]. The GLOBOCAN data indicated that the frequency of occurrence depended on the region and population type and that certain types of cancers are more commonly diagnosed than others. These include lung cancer, which is more prevalent among tobacco smokers, breast cancer diagnosed in both men and women, and colorectal cancers, including colon and rectal cancers, which are particularly prevalent in Western societies. In addition, prostate cancer and liver cancers, especially those related to hepatitis B and C viruses, vary considerably. The incidence of stomach cancer is also decreasing in many areas but can still constitute a significant social problem. Finally, cervical cancer is very much dependent on human papillomavirus (HPV) infection.

One of the most commonly diagnosed cancers worldwide in 2020, according to GLOBOCAN, was non-melanoma of the skin (excluding basal cell carcinoma), which was identified in a total of 1,198,073 patients (722,348 cases in men and 475,725 in women), i.e., 6.2% of all cancer cases. The number of deaths involving all types of skin cancers other than melanoma was 63,731 (0.6%). Moreover, during this period, 173,844 men and 150,791 women were diagnosed with melanoma. Melanoma was the cause of death for over 57 thousand individuals [1]. Although studies have revealed large geographical differences in the incidence of melanoma according to country and region, melanoma is generally more common in men than women [2].

Epidemiological data often does not discriminate between all skin cancer subtypes, such as basal cell carcinoma, and as such, significant discrepancies exist regarding the numbers of newly diagnosed cases. In addition, many tumors are quite common but often have a benign course, leading to their treatment within primary healthcare settings without formal reporting. Also, as many benign changes are underdiagnosed, the number reported in national cancer registries is likely significantly lower than the true extent. The incidence of skin cancer has undoubtedly been increasing in recent decades, with most of these changes being associated with repeated exposure to sunlight, climate change, including changes in the ozone layer, and changes in individual and social habits [3]. The reports of the World Cancer Research Fund International (WCRFI) and the World Health Organization (WHO) on skin cancer statistics and the forecast of future cases are disturbing (Figure 1) [4,5,6,7].

## 2. Skin Cancers

Skin cancers most often are divided into two groups: non-melanoma (NMSC) and melanoma (*malignant melanoma*; MM) [8]. The NMSCs can be distinguished into basal cell carcinoma (*carcinoma basocellulare*; BCC) and squamous cell carcinoma (carcinoma spinocellulare; SCC); Figure 2. The most common skin cancer is BCC which accounts for 80% of all skin malignancies (ratio of incidence compare with SCC is between 10:1 and 1:1) [9]. MM accounts for 1.5–2% of all skin cancers [10,11].

Basal cell carcinoma is characterized by slow growth and slight and locally located malignancy [9]. The development of BCC is favored by precancerous conditions or previously unchanged skin and exposure to UV radiation (280–320 nm), but BCC can also affect the fatty tissue under the skin or spread even more. The highest rate of morbidity is observed in people over 65 years of age, which is more than 95% of all basal cell carcinomas [12]. Potential tumor-promoting factors include solar radiation (UVB), long-term exposure to exposed body parts, arsenic-type chemicals, soot, HPV viruses, and X-rays [9,13]. Precancerous conditions include actinic/actinic keratosis (keratosis senilis / actinic) with a risk of transformation, Xeroderma pigmentosum as a genetic defect (increasing the incidence of skin neoplasms), radiation dermatitis (radiodermatitis), as well as chronic inflammation, scarring or hypertrophic burn scars [14,15,16,17]. Squamous cell carcinoma develops from flat squamous cells that form a significant proportion of the epidermis (keratinocytes), the outermost layer of the skin [18,19,20]. SCC may develop in previously unchanged skin or within the focus of senile keratosis, white keratosis (leukoplakia), in burn scars, as well as within chronic ulcers [21]. Squamous cell carcinomas usually grow slowly and often spread or metastasize. Basal cell carcinoma can also affect the fatty tissue under the skin or spread even more.

Melanoma is one of the most aggressive cutaneous malignancies and it was first described in 1812 by Rene Laennac [22]. The tumor originates from the malignant transformation of pigment cells, i.e., melanocytes responsible for melanin synthesis [23]. Usually, melanoma is formed de novo but can develop based on damaged skin or skin lesions (pigmented nevi). The incidence of melanoma has increased dramatically in recent years. Over the past fifty years, there has been a five-fold increase in the incidence of melanoma among pale-skinned people [24].

## 3. Photoaging as a Risk Factor for Skin Cancer Development

The skin undergoes various age-related changes, including the loss of collagen, elastin, and other structural components, resulting in the appearance of inter alia wrinkles, loss of firmness, and pigmentation changes. Various endo- and exogenous factors contribute to skin aging, including exposure to UV radiation (UVR), genetics, lifestyle, and diet [25], as well as a number of environmental factors, such as air pollution, tobacco smoke, changing climatic conditions, and ultraviolet radiation [26]; ultraviolet radiation itself is 80% responsible for skin aging. Exposure results in photoaging, leading to tissue damage, dysfunction, and skin structure [27]. Skin can be aged by both natural and artificial radiation sources [28]. Either way, photoaging alters the composition of the extracellular matrix, suppresses the immune response and induces immunotolerance [29]. These changes can lead to the development of skin cancers and precancerous conditions [27,30,31,32,33,34,35].

Photoaging and the formation of lesions depend mainly on the time of exposure to UV radiation (UVR), the occurrence of burns, and the skin phototype. Carcinogenesis is most commonly associated with phototypes I and II (light skin, blond hair, a tendency to sunburn, freckles, and numerous nevi). In addition, short but frequent periods of radiation exposure have a higher risk of causing cancer, especially in light-skinned people not accustomed to high doses of UVR, and melanoma is more common in adults who were exposed to high doses of sunlight in childhood [19]. Epidemiological studies towards the end of the 20th century indicate a higher likelihood of melanoma among adults in Australia who moved from the UK while they were children [31].

Most of the UVR reaching the earth consists of UVA (95%) with some UVB (5%) [7,32,34]. Wavelengths below 290 nm are blocked by the ozone layer; however, higher levels of harmful radiation have been observed under ozone holes, e.g., in Australia [36]. A loss of 1% of the ozone layer causes an increase of 1–2% of incidence of melanoma [37]. UVA radiation reaches the deep dermis, up to fibroblasts and germ cells of melanocytes and epidermal keratinocytes [38], with over 50% penetrating the reticular and papillary layers of the skin.

UVA radiation can cause photoallergic and phototoxic reactions, and encourage free radical formation; it can also damage structural proteins and DNA, resulting in carcinogenic and mutagenic properties [39]. UVB radiation promotes the depletion of Langerhans cells in the epidermis and causes photoallergic reactions [40]. They can damage the DNA in skin cells directly and are the main rays that cause sunburns and erythema and can lead to irritation of the cornea and conjunctiva and even cataracts. They are also thought to cause most skin cancers [41,42,43,44,45,46].

The International Agency for Research on Cancer (IARC) has recognized natural UVR as the main factor responsible for cancer development. The effects of UVR are not always negative. Shorter exposure and lower absorbed doses promote positive photochemical reactions, such as the synthesis of vitamin D or melanin, which is a natural protective factor against UV [35,38]. During exposure to sunlight, UVB photons photolyze the 7-dehydrocholesterol into provitamin D_3_, which is isomerized to cholecalciferol (vitamin D_3_) [39].

## 4. Molecular Basis of Melanoma Development

Melanoma is largely associated with a genetic predisposition. Studies indicate that approximately 10–15% of all melanoma patients have hereditary melanoma associated with the presence of a mutation in a single high-risk cancer predisposition gene, *CKN2A*, sun exposure, or an identical skin phototype [47,48]. This genetic change is driven by disorders in the suppressor genes that inhibit cell division, proto-oncogenes that activate cell proliferation, as well as MHC—immune control genes that oversee the repair of mutations in the human genome related to angiogenesis processes [30].

In 1994, a mutation of the CDKN2a/MTS1/INK4A gene located on chromosome 9p21 was found in patients with melanoma [42]. The gene belongs to the group of suppressors and is responsible for the coding of two proteins: p16 (inhibits pRb phosphorylation) and p14, which increases p53 activity by binding to MDM2. Both pRb and p53 proteins, due to their inhibitory effect on the cell cycle, play an important role in the process of cell apoptosis [49]. When pRb mutations occur, cells with damaged genetic material begin to divide uncontrollably.

Mutations in the CDKN2a/p16 gene are responsible for 25% of familial melanomas [50]. It has also been shown that almost 8% of patients with multiple melanoma foci have mutations in *INK4a*, while no such changes were noted in the rest. The mutation in the *INK4a* was also assessed in patients with melanoma for the first time [51]. The CDK14 proto-oncogene located on chromosome 12q13 in the mutated form turns into an active oncogene. In vitro studies of melanoma cells have identified an R24C mutation in CDK4 [52].

Inherited mutations can significantly shorten the time needed for tumor development. The accumulation of mutations is influenced by both external and internal factors, the effectiveness of the detoxification systems, and the efficiency of DNA damage repair mechanisms. Notably, UV radiation is among the most potent external mutagenic factors (Figure 3).

The formation of melanomas under the influence of UVR is a complex process involving a series of molecular mechanisms, including mutagenic action, damage to pathways leading to apoptosis, and the promotion of the proliferation of altered, immature cells (Figure 4) [54]. UVB radiation, i.e., with a shorter wavelength, can directly damage the DNA of skin cells by inducing pyrimidine dimers, most commonly cyclobutane thymine dimers (CPD) and (6-4) photoproducts. This physical damage to the DNA structure poses a direct threat to gene integrity, leading to the occurrence of mutations. In turn, mutations affect the expression of proteins, such as p16INK4a; this is an inhibitor of cyclin-dependent kinases (CDK4, CDK6), and serves as a critical checkpoint allowing the cell to transition from the G1 phase to the S phase, preceding mitotic cell division. It also influences p14ARF proteins, which participate in the degradation of p53 and increase its activity, by binding to the MDM2 protein.

Activated p53 is a key tumor suppressor protein. It plays a role in repairing DNA damage, or induces apoptosis if the damage is too severe to repair [55,56,57]. In vitro tests have found UVB radiation to arrest human melanocytes in the G1 phase, which is associated with prolonged expression of p53 and p21 [58,59]. It has also been observed that UVR affects the activation of the MAPK (mitogen-activated protein kinase) pathway through mutations in the BRAF gene and changes in the expression of RAF proteins. This results in the activation of the MAPK pathway, leading to uncontrolled cell division and melanoma progression. Furthermore, one of the characteristics that distinguishes melanoma from normal melanocytes is a change in the expression of cadherins, i.e., adhesion molecules involved in cell-to-cell interactions, and contribute to the formation of metastases [60]. Meanwhile, in vitro studies have shown that UVB radiation induces the secretion of endothelin 1 (ET-1) by keratinocytes, causing a reduction in E-cadherin production by influencing melanocytes and melanoma cells [61].
Figure 4Molecular mechanisms of the influence of UV radiation on the development and progression of melanoma. (The scheme was prepared based on [52,62]).
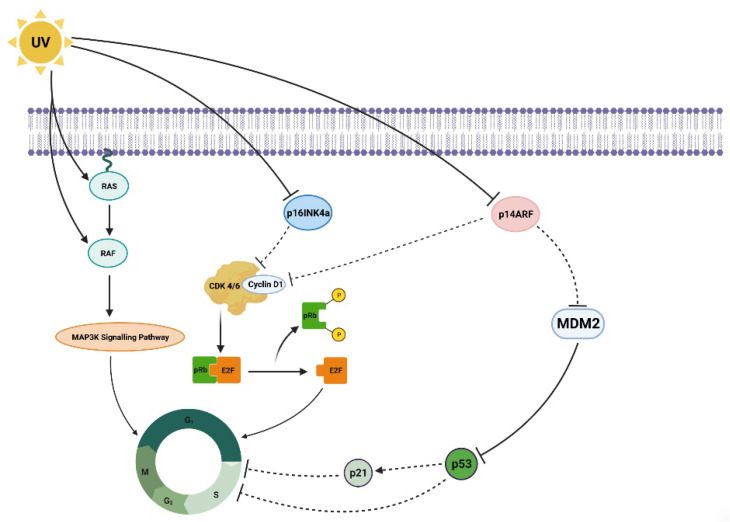


## 5. Nanotechnology as an Innovative Approach to Diagnosing and Treating Skin Cancers

The fight against cancer must be supported to improve overall health in both Europe and elsewhere. Most current research aims at the early detection and effective treatment of oncological diseases, although some efforts are also made to improve the quality of life of patients, e.g., by pharmacologically reducing pain, which is often an inherent element of the disease. Despite the enormous progress of modern medicine and the dynamic development of pharmacology and intensive scientific research, no fully effective, minimally invasive anticancer chemotherapy currently exists. The clinician must choose from a range of cytostatic drugs, each with a unique mechanism of action and application in the treatment of specific types of tumors, e.g., alkylating cytostatic drugs that disrupt the DNA structure of cancer cells (e.g., cyclophosphamide, cisplatin), antimetabolites that affect nucleic acid synthesis (5-fluorouracil [5-FU], methotrexate), microtubule-targeting drugs that prevent normal cell division (paclitaxel), and anthracycline antibiotics that disrupt DNA function (doxorubicin). Unfortunately, due to their non-specific activity towards cancer cells, many anticancer therapies have undesirable side effects and must be administered at high doses which are toxic to healthy tissues. Moreover, these drugs are often used in various combinations, depending on the type of tumor and its response to treatment. The most serious complications of chemotherapy are cardiotoxicity, hepatotoxicity, neurotoxicity, and nephrotoxicity, which may appear even many years after the end of therapy. Therefore, a great challenge for modern oncology is the development and synthesis of effective yet safe anticancer drugs with very low, or no, systemic cytotoxicity against normal cells, no side effects during and after the therapy, high specificity of action against cancer cells, and rapid elimination from the body in the least burdening way, and which are characterized by a lack of immunogenicity and mutagenicity [63,64,65].

Great hopes are currently associated with nanotechnology, a new interdisciplinary branch of science and technology dealing with the design and creation of structures called nanoparticles, which range in size from 5 to 100 nm. It is currently one of the most popular fields of science and its development is of great importance in pharmacy and medicine. In recent years, nanotechnology has been of particular interest due to the high anticancer potential, relatively high durability, and low cytotoxicity of nanoparticles to normal cells. Recent development in nanotechnology provide the opportunity to effectively treat cancer by increasing the bioavailability, targeting, and delivery of drugs at effective concentrations to cancer cells; such developments also avoid the phenomenon of drug resistance [63,64,65].

Current methods of treating skin cancers include photodynamic therapy, radiotherapy, and surgical excision of the tumor with a margin of healthy tissue. Conventional chemotherapy plays a limited role in treatment, especially in the case of melanoma. Occasionally, it is used after the surgical removal of the tumor to eliminate any potential cancer cell remnants, and in advanced stages of melanoma when the cancer has spread to other organs. Unfortunately, the effectiveness of chemotherapy in melanoma is limited, leading to the inclusion of alternative treatment methods such as immunotherapy and molecular and targeted therapies. Chemotherapy is used slightly more often in the treatment of other types of skin cancers, such as basal cell carcinoma or squamous cell carcinoma, especially in advanced cases [64,66].

A growing number of in vitro and in vivo studies suggest that nanotechnology may be effective in the treatment of skin cancer. Nanomaterials and nanocarriers allow the development of drug delivery systems with greater biological effectiveness at lower doses and lower side effects. Due to the small size and surface characteristics of nanomaterials, the anticancer drugs loaded into such nanoparticles easily penetrate cell membranes and can be delivered directly and specifically to skin cancer cells, where they can then exhibit maximum effect. In addition, due to the increased activity of anticancer drugs administered by nanocarriers, the systemic side effects of chemotherapy are reduced; anticancer drugs that demonstrate greater effectiveness at lower doses play a key role in improving the overall health of patients. It should also be emphasized that the use of appropriate nanocarriers can counteract the biodegradation of cancer drugs in the patient’s body, reduce their removal from cells, or extend their half-life [67,68,69].

Various nanostructured platforms, including liposomes, carbon nanotubes, nanomicelles, nanoemulsions, and metal nanoparticles have been explored for their potential in enhancing skin cancer diagnosis and treatment. These nanosystems offer unique advantages in targeting tumor cells, enhancing drug delivery, and improving therapeutic efficacy (Figure 5).

### 5.1. Lipid-Based Nanoparticles in Skin Cancer

Lipid formulations have emerged as a promising technology to enhance drug efficacy and safety. By selectively delivering drugs to specific cells and organelles, lipid formulations can reduce side effects and improve treatment outcomes. Additionally, lipid formulations can stabilize drugs, making them easier to store and transport [74]. Among the various lipid formulations, lipid nanoparticles are particularly promising due to their unique composition and structure. These spherical vesicles can solubilize and deliver drugs efficiently, improving their bioavailability and therapeutic potential [75]. Lipid-based nanoparticles are considered to be one of the most suitable carrier systems for therapeutic compounds due to their unique lipid composition [76]. Lipid nanoparticles are a diverse group, including liposomes [77], ethosomes [78], niosomes [79], and solid lipid nanoparticles [80], each with its unique properties and applications. Despite this, extensive research has been conducted on these compounds to evaluate their effectiveness in delivering anticancer therapies for various cancers, including skin cancer, while minimizing side effects.

#### 5.1.1. Liposomes

Liposomes have become a universal tool in biology, biochemistry and pharmacy due to their great structural diversity. They are made of a lipid bilayer surrounding a central aqueous space containing the transported drug or other bioactive molecule [69,81,82]. The use of liposomes enhances the biological effectiveness of routinely administered chemotherapeutic agents at significantly lower concentrations. This provides hope for patients in reducing the unfavorable side effects commonly observed during chemotherapy. This can be attributed to improvements in pharmacokinetic parameters, such as better drug/biomolecule delivery to cancer cells and overcoming both hydrophobic and hydrophilic barriers [83].

The fact that liposomes are rapidly taken up by the reticuloendothelial system and degraded by macrophages is an undeniable limitation. Negative aspects associated with their use as drug carriers can be eliminated by modifying their surface, for example, by introducing PEG polymers or cationic lipids into their structure. These chemical modifications effectively enhance the pharmacokinetic and pharmacodynamic properties of the liposomes. It is important to note that one of the most commonly used methods today is to coat the surface of the liposome with polyethylene glycol (PEG), creating what is known as long-circulating liposomes or ‘stealth liposomes’. Obviously, it is possible to further modify the surface of the liposomes by incorporating various ligands onto the surface of the liposome, such as glycoproteins, immunoglobulins, peptides, transferrin, etc., in order to preferentially target them to overexpressed receptors in tumor cells [84].

Many strategies have been trialed for utilizing liposomes in the treatment of skin cancer. These involve both the use of routinely administered chemotherapeutic agents (e.g., doxorubicin or fluorouracil) [85,86,87], natural polyphenols as chemotherapy adjuvants (resveratrol, curcumin, epigallocatechin) [87,88,89,90,91], and gene therapies, such as the application of siRNAs to modulate the expression of specific genes of cancer cells [92,93,94].

Encapsulation of doxorubicin, paclitaxel, or 5-fluorouracil improves their pharmacokinetics, thus increasing their half-life in cancer cells [84,85,86]. Sing et al. [86] showed that the encapsulation of DOX and celecoxib (CEL), a non-steroidal anti-inflammatory drug, in liposomes significantly increased the biological activity of chemotherapeutic agents in human skin carcinoma A431 cell culture. These dual drug-loaded liposomes were able to inhibit cancer cell viability by up to >99%, even at lower concentrations. The co-exposure of doxorubicin and celecoxib synergistically inhibited the AKT and COX-2 pathways leading to cell apoptosis [86].

The progression and metastasis of skin cancers is often associated with the overexpression of receptors for growth factors, such as the epidermal growth factor receptor EGFR on the surface of tumor cells. Some of the most promising nanocarriers for targeted drug delivery in skin cancer treatment are receptor-specific liposomes, conjugated with appropriate ligands. This technique has been combined with the use of aptamers due to their ability to recognize antigens, ease of chemical variability, and sequence changes. One study examined the drug release kinetics, in vitro cell viability, in vitro targeting capability, and apoptotic effects of a combination of AS1411 aptamer-functionalized liposomes loaded with 5-FU on human dermal fibroblasts (HDF) and the BCC cell line TE 354.T [87]. It has been shown that aptamer conjugation increased liposome size and reduced the surface potential of the liposomes; moreover, the aptamer moieties increased the stability of the liposomes and acted as a supplementary steric barrier leading to a lower cumulative amount of the released 5-FU. The results indicate that aptamer conjugation increased liposome size and reduced the surface potential of the liposomes, and that the aptamer moieties increased the stability of the liposomes and acted as a supplementary steric barrier, reducing the cumulative amount of the released 5-FU. The results indicate that the functionalized liposomes are more efficient as nanocarriers than the non-functionalized ones. The therapeutic effect of 5-FU was also improved by eliminating a number of secondary, undesirable effects which accompany the classic one-drug administration.

Another study found the use of 5-FU aptamers to significantly increase its specificity against cancer cells [87]. AS1411 aptamer-functionalized liposomes loaded with 5-FU may have potential as effective and targeted treatment in basal cell carcinoma (BCC). The use of the AS1411 aptamer is significant because it specifically binds to nucleolin, a protein present in the cell membrane of various cancer cells, including basal cell carcinoma (BCC).

Petrilli et al. [85] evaluated the potential of EGFR targeted immuno-liposomes, composed of cetuximab encapsulated by 5-FU, against squamous cell carcinoma (SCC) in vitro using A431 (EGFR positive) and B16F10 (EGFR negative) cell lines, as well as in an in vivo animal model. The cells are characterized by significant overexpression of the EGRF receptor. The liposomes, consisting of an antibody and a chemotherapeutic agent, demonstrated better biological activity than in the case of 5-FU used alone.

The interaction between surface receptors and specific ligands is crucial for skin cancer progression and cell proliferation [95]. One classic method of eliminating CD20+ melanoma stem cells responsible for tumor initiation and metastasis is based on the interaction between active molecules and the receptor. Zeng et al. [96] found salinomycin-loaded lipid-polymer nanoparticles with anti-CD20 aptamers to be significantly more effective in inhibiting mouse tumor cell proliferation than salinomycin alone.

Currently, great expectations in cancer treatment are associated with gene therapy and the use of miRNA, siRNA, and shRNA. Appropriate modulation of genes associated with the proliferation of cancer cells can inhibit cancer progression, block the cell cycle and ultimately cause cell death, e.g., by apoptosis. Numerous studies indicate that the use of siRNA particles encapsulated with liposomes yields very good results in the treatment of skin cancer. The main targets for this therapy comprise the genes/pathways responsible for the proliferation of cancer cells. Interesting results were obtained for siRNAs inhibiting the Akt and MAPK/PI3K pathways and c-Myc gene activity; the suppression of c-Myc production in tumors inhibited tumor progression in mouse models [93,95,97].

In vitro and in vivo studies conducted by Jose et al. found co-encapsulated curcumin and anti-STAT3 (signal transducer and activator of transcription 3) siRNA using cationic charged liposomes to be effective against skin melanoma [91,92]. A series of studies of B16F10 and A431 murine melanoma cells showed that liposomes containing STAT3 siRNA effectively inhibited cell proliferation, and induced death by apoptosis.

Another strategy in the use of liposomes in the treatment of skin cancer is the encapsulation of natural polyphenols, such as resveratrol, quercetin, or epigallocatechin. Encapsulation of polyphenols in liposomes causes a significant increase in their cellular uptake by cancer cells compared to their use as a single agent. These compounds act as effective scavengers of the high numbers of free radicals generated in cancer cells. Studies have found that the use of polyphenolic liposomes led to a significant reduction in inflammation in skin cancer cells [85].

Classical photodynamic therapy (PDT) often suffers from limited effectiveness in treating cancer and severe phototoxicity after treatment, resulting in light-induced skin damage. However, it has been proposed that using nanotechnology as carriers of photosensitizers increases the effectiveness of PDT in cancer treatment and improves patient comfort [98]. There are increasing hopes for using liposomes in modern photodynamic therapies in treating skin cancer; their use as carriers of photosensitizing compounds may improve treatment effectiveness.

Notably, as outlined by Feng et al. [99], the conjugation of chlorin e6 (hCe6) as photosensitizer together with a lipophilic near-infrared (NIR) dye 1,1’-dioctadecyl-3,3,3′,3’ -tetramethylindotricarbocyanine iodide into liposomes brought very good results. In vivo studies have demonstrated that the use of liposomal encapsulation significantly enhances the effectiveness of skin cancer treatment with excellent biocompatibility. Liposomal encapsulation has been shown to enhance the selectivity of the therapy towards cancer cells while maintaining excellent biocompatibility [99]. According to the study authors, liposomes can enhance safety and comfort for patients during PDT. Furthermore, recent research by Pivetta et al. [100], using liposomes containing methylene blue and acridine orange, demonstrated a significant improvement in the effectiveness of their biological action when combined with photosynthetic agents and nanoparticles. A significant rise in the phototoxicity potential of photosensitizer liposomes was observed at extremely low concentrations.

There are several strategies for using liposomes in the treatment of skin cancer. For instance, liposomes can encapsulate the human gene of the interferon B protein [101] or the UV-DNA repair enzyme T4N5 [102], which have higher anticancer/preventive effectiveness than molecules used alone. In addition, the liposomes may have applications in the development of vaccines for the treatment and prevention of melanoma. Subsequent studies indicate their potential role in immunotherapy against melanomas [101,102,103,104,105].

#### 5.1.2. Ethosomes

When discussing lipid nanoparticles, it is important to mention the increasing popularity of ethosomes. Ethosomes are an area of rapidly growing research, with studies investigating their potential to treat a wide range of diseases. This nanocarrier system shows promise for various medical applications. Ethosomes, which contain ethanol in their formulation, have unique properties that distinguish them from liposomes. Due to the presence of alcohol, it has been found that there are a number of unique properties that can improve the efficacy and safety of drug delivery [106]. Ethosomes are a promising but relatively unexplored delivery system that holds immense potential for improving melanoma treatment. Recent studies have demonstrated their ability to effectively penetrate the skin and selectively deliver drugs to melanoma cells, leading to enhanced therapeutic outcomes. However, more extensive clinical trials are needed to firmly establish the safety and efficacy of ethosomes as a melanoma treatment modality [107,108,109,110,111].

Several recent studies, both in vitro and in vivo, have demonstrated that combining ethosomes with classical chemotherapy and drugs can be an effective treatment. Khan and Wong’s study showed that encapsulating 5-FU into ethosomes significantly increased drug penetration into the skin and retention, resulting in increased effectiveness of the chemotherapy drug [112]. In relation to this study, there are also results from another team of researchers who have used a combination of mitoxantrone and ethosomes. They observed a higher permeability of nanoparticles through the skin of rats and a significantly higher in vivo antimelanoma effect than MTO solutions [113]. It is important to note that the enhanced efficacy of drugs when combined with ethosomes is not limited to traditional chemotherapeutic agents like 5-FU or paclitaxel. In recent research, Mousa et al. [114] achieved compelling in vitro and in vivo results with encapsulated metformin. The compound inhibits the growth of skin cancers in vitro. However, the use of appropriate ethosomes significantly increased the antitumor activity against skin cancer compared to the application of free metformin in male Swiss albino mice.

It is important to note that satisfactory results have been obtained in studies using a combination of ethosomes and natural compounds that not only have cytotoxic properties, but also contribute to redox homeostasis and inhibit the generation of free radicals in skin cancer cells [110,111,115]. Studies conducted by independent research teams using compounds such as epigallocatechin [111], nobiletin [110], or fisetin [115] indicate that the combination of natural compounds with ethosomes increased their activity in skin cancer cells compared to non-encapsulated solutions of phytochemicals. The authors emphasize that better anticancer effectiveness results from the increased availability of phytochemicals and their better accumulation directly in the environment of skin cancer cells. Histopathological analyses conducted in in vivo studies showed a reduction in tumor size in mice after the administration of nanoparticles. Furthermore, biochemical quantification of oxidative stress biomarkers, such as glutathione, superoxide dismutase, and catalase, indicated better inhibition of reactive oxygen species generation in skin cancer cells treated with phytochemicals encapsulated in ethosomes [110,111,115].

#### 5.1.3. Solid Lipid Nanoparticles

Solid lipid nanoparticles (SNLs) are colloidal lipid carriers with a typical size range of 50–1000 nm. The SLNs are composed of natural lipids, including fatty acids, steroids, waxes, monoglycerides, diglycerides, and triglycerides. The solid lipid core matrix of SLNs encapsulates lipophilic or hydrophilic drugs, depending on the preparation method. Surfactants are used to stabilize the core lipid matrix. However, their ability to encapsulate anticancer agents and safely transport them to the tumor site for controlled release, without causing any permeability or toxicity issues, has made them the most competitive drug carriers for skin cancer therapy [116].

According to Kim et al.’s [117] research, the encapsulation of docetaxel into SNLs led to a significant improvement in the drug’s biological activity against melanoma. It inhibited growth and prevented tumor formation in mice, which was significantly superior to the administration of free docetaxel. Additionally, the treatment resulted in an increase in the population of cytotoxic T cells, while the population of tumor-associated macrophages and regulatory T cells decreased [117].

In line with these reports are also recent studies using 5-FU [118,119] and dacarbazine [112]. The use of SNLs for drug encapsulation resulted in a significant improvement in their anticancer properties, increased drug retention and bioavailability. Histopathological analysis showed that rats treated with decarbazine-SNLs had less keratosis, inflammatory responses, and angiogenesis than rats treated with free dacarbazine [120]. Similar effects were observed in independent studies in in vitro [119] and in vivo [120] experiments. Mice treated with 5-FU-SNL exhibited decreased inflammatory responses, less keratinization, and reduced signs of angiogenesis when compared to mice treated with 5-FU [120].

### 5.2. Inorganic Nanoparticles

Inorganic nanoparticles have shown considerable potential in the fight against cancer. They can deliver drugs directly to cancer cells, image cancer lesions, and enhance the effects of radiotherapy. Nanoparticles can be produced from various materials, including metals and their oxides, carbon, or silica. Their unique properties, such as small size, large surface area, bioactivity, biocompatibility, and modifiability, make them ideal candidates for skin cancer therapy. Various strategies exist for utilizing inorganic nanoparticles in anticancer therapy. While some molecules possess antiproliferative properties, they are also used as effective drug carriers or as photosensitizers in classical photodynamic therapies [121,122].

#### 5.2.1. Functionalized Metal Nanoparticles

Currently, gold and silver nanoparticles are most often used in the treatment and therapy of skin cancer and have been thoroughly tested in recent years. Due to their small size, these molecules easily penetrate healthy cells and accumulate in cancer cells, ensuring high concentrations of chemotherapeutic agents in cancerous cells. Gold and silver nanoparticles are used for administering targeted medication, monitoring tumor progress, vaccinations, and as potent chemical sensors or as elements of therapy combined with photodynamic therapy (PDT) [123,124,125,126]. Gold nanoparticles are of particular interest due to their potential to increase the activity of anticancer chemotherapeutic agents while reducing the side effects of treatment by enabling lower doses [127,128]. One of the main strategies in the use of metal nanoparticles is their combination with available chemotherapeutic agents. The latest research shows that the use of gold [129] and silver [130] nanocarriers in combination with 5-FU significantly increases its anticancer activity. Greater cytotoxicity of the drug was observed in relation to conventional chemotherapy. It was shown that metal nanocarriers increased drug stability and clearly improved its pharmacokinetics in cancer cells. The higher concentration of 5-FU in cancer cells and the targetability resulted from a much better penetration of cancer cells by nanoparticles than in the case of 5-FU used alone [129,130].

Preet et al. evaluated the effects of synthesized gold nanoparticles loaded with nisin and doxorubicin, as a combined approach to fight murine skin cancer [131]. The results obtained were highly satisfactory. A significant decrease in tumor cell proliferation was observed compared to controls, and the nanoparticles demonstrated immunomodulatory properties: treatment was associated with a decrease in serum pro-inflammatory cytokines, including tumor necrosis factor TNF alpha and beta, nuclear factor kappa B (NF-kB), and interleukins 1 and 10 (IL-1, IL-10). This resulted in apoptosis, probably due to, inter alia, the generation of ROS in cancer cells and lipid peroxidation.

Gold nanoparticles encapsulated with methotrexate also yielded good results against moderate to severe inflammatory diseases, as noted on in vivo and in vitro skin models [132]. Topical treatment with AuNPs-3MPS@MTX reduced keratinocyte hyperproliferation, epidermal thickness, and inflammatory infiltration in vivo in a mouse model of imiquimod-induced psoriasis.

A promising approach is to combine natural phytochemicals with metal nanoparticles of gold or silver. Combining polyphenols with nanoparticles has been found to yield synergistic anticancer properties. Numerous studies using curcumin [133], *Vitis vinifera* [134], *Siberian ginseng* [135] or *Water Chestnut* [136] indicate that nanoparticles significantly contribute to improving the antiproliferative properties of phytochemicals against skin cancer. As in the case of chemotherapeutics, such improvements have been attributed to enhanced distribution and release of polyphenols in cancer cells. Importantly, such combinations often induce apoptosis in skin cancer cells. The studies showed an increase in the activity of anti-apoptotic BH3-only proteins from the Bcl-2 family and a simultaneous decrease in their anti-apoptotic partners. It is believed that these pro-apoptotic properties are based on the pro-oxidative activity of the compounds and the generation of significant amounts of ROS in cancer cells. Modern therapies often use radiotherapy to increase the effectiveness of metal nanoparticles. PDT (photodynamic therapy) has been found to be particularly effective, and is becoming increasingly popular for treating various types of cancer, including skin cancer [137]. However, the organic photosensitizers used in PDT are often burdened with numerous disadvantages, such as high systemic toxicity, low selectivity of action towards cancer cells or low level of light absorption.

Noble metal nanoparticles are characterized by high chemical and physical stability, minimal toxicity to normal cells, and high selectivity of action, and are gaining popularity as photosensitizers. The latest research shows that gold [138] and silver [139] nanoparticles can be successfully used in modern photodynamic therapies. The molecules are characterized by good biocompatibility and bioavailability, with a clear accumulation in the tumor, which additionally significantly improves the effectiveness of photothermal therapy (PTT) or PDT in the treatment of melanoma. Xie et al. [138] report that gold nanoparticles not only yielded a potent PTT/PDT effect on destroying the primary tumors, but also elicited strong antitumor immunity for eliminating primary and metastatic melanoma; they can also relieve immune suppression by promoting T cell infiltration into tumors, and maintain lasting anti-tumor immunity for long-term prevention of melanoma recurrence.

Metal nanoparticles have also been used to support biopsy and radiotherapy, which may not be sensitive enough to detect melanoma at an early stage. Surface-enhanced Raman spectroscopy (SERS) is gaining popularity in bioimaging and diagnostics. Au NPs (nanoparticles) are considered excellent for in vivo imaging applications because they are inert, biocompatible, and their localized surface plasmon resonances (LSPR) can be aligned towards near infrared (NIR) regions [140,141].

#### 5.2.2. Carbon Nanotubes

Carbon nanotubes are characterized by a unique structure with interesting optical, chemical, physical, and mechanical properties. One of the advantages of nanotubes is their ability to penetrate cell membranes and carry small molecules or biological macromolecules such as plasmids, siRNA, or proteins into cells. They can be used as a biomarker sensor for the diagnosis of skin melanoma and infection at an early stage. Moreover, carbon nanotubes offer targeted delivery to the cancerous cells, act selectively, and provide better penetration in the neoplastic cells due to improved permeability and retention effect [142,143,144]. Several studies indicate that the use of carbon nanotubes improves the effect of drugs and affects the chemical stabilization of chemotherapeutic agents [145]. Sahoo et al. [146] found carbon nanotubes combined with graphene oxide loaded with the anticancer drug camptothecin to effectively inhibit the proliferation of breast and skin cancer cells.

#### 5.2.3. Nanofibers

The use of nanofibers in treatment can be very diverse, and as research shows, they can be a good transporter for both natural compounds and synthetic chemotherapeutics [147]. Rengifo et al. [148] report that the use of appropriate nanofibers in combination with an anticancer compound may not only increase its cytotoxicity. Studies on B16F10 melanoma cells showed that nanoparticle delivery significantly improved control of drug release in local chemotherapy of skin cancer. Nanoencapsulation increased both skin compound permeation and retention. A recent study by Balashanmugam et al. [149] found polymers composed of phytosynthesized AuNPs and curcumin for the treatment of skin cancer A431 cell to exhibit selective toxicity; nanofiber treatment induced apoptotic death in cancer cells but not in normal cells.

Nanofibers are also successfully used in combination with commercially used drugs, e.g., metformin [150], or commonly used chemotherapeutics, such as 5-FU [151], etoposide, and methotrexate [152]. The use of metformin surface modified cellulose nanofiber gel resulted in a significant decrease in the invasiveness of murine melanoma cell B16F10. Treatment yielded high suppression of skin cancer cell migration and a significant inhibition of their proliferation and growth. This indicates that the strategy is a promising approach for preventing melanoma metastases [150].

Extremely valuable results were achieved regarding the use of cellulose nanofibers modified with Fe_3_O_4_-Ag_2_O quantum dots as a carrier of anticancer drugs for skin cancer [152]. Importantly, while these compounds exhibited very low cytotoxicity against normal cells, they also demonstrated considerable potential against skin cancer cells. The drug was found to have greater anticancer potential and cytotoxicity against the human melanoma SK-MEL-3 cell line, which may have been due to its selective release.

Zhu et al. [151] report that the use of appropriate, novel core-shell nanofibers based on chitosan (CS)-loaded poly (ε-caprolactone) and a 5-fluorouracil (5-FU)-loaded Poly(N-vinyl-2-pyrrolidone) (PVP) core increased the anticancer activity of the chemotherapeutic agent against melanoma skin cancer cells (B16F10 cells). The nanoparticle showed significant inhibitory proliferation effects on B16F10 cells in vitro through arresting cell cycle progression at S phase and G2/M phase in time-dependent manner. More importantly, 5-FU in this form showed significantly less activity against normal cells. This data hints at a promising future cancer treatment strategy, and the potential for synergism may expand the possibilities of designing chemotherapy therapy with minimal adverse effects on normal cells.

### 5.3. Polymer-Based Nanoparticles

Polymer-based nanoparticles are drug carriers made from synthetic or natural polymers. They are divided into different types based on their shape and the properties of the polymer used, such as micelles, dendrimers, polymersomes, and polyplexes. These nanoparticles have several advantages, including improved preparation techniques, biocompatibility, biodegradability, and lower production costs. From a biological perspective, polymer-based molecules possess several advantageous characteristics. These nanoparticles can conjugate, adsorb, capture, or encapsulate anticancer agents, including hydrophilic and lipophilic drugs, monoclonal antibodies, and genes, among others, for controlled release, tumor targeting (active/passive), protection under physiological conditions, and enhanced tumor uptake [153].

#### 5.3.1. Functionalized Polymeric Nanoparticles

Polymer nanoparticles can increase the efficiency of transport of used drugs and proteins to target cells to reduce their toxic effects. Their nanoscale size allows them to effectively penetrate cell membranes and increase their stability, which allows the drug to stay in circulation longer [154]. The anticancer effects of most melanoma anticancer drugs are limited due to their lipophilic structure and hence unfavorable pharmacokinetic and pharmacodynamic properties. However, the free drug release profile of anticancer drug formulations has been improved by the use of amphiphilic polymers, i.e., those with both hydrophobic and hydrophilic sections [155].

Polymer nanoparticles offer promise due to their greater stability, controlled release, and enhanced skin permeation. Different forms of polymer nanoparticles, such as nanospheres and nanocapsules, polymer micelles, dendrimer-based micelles, and polymer-drug conjugates, can be produced by altering the properties of the polymer [156,157]. Zou et al. [158] report that, like liposome systems, polymer nanosystems can not only significantly improve CT scan imaging of tumors, but also mediate effective targeted chemotherapy for melanoma.

Natural polymeric nanoparticles like chitosan, gelatin, albumin, and alginate are most frequently used for topical skin delivery and targeting skin melanoma. These compounds are often characterized by high chemical stability and good penetration of skin cells and possess very valuable antioxidant, antibacterial, and anti-inflammatory properties [159].

There are many strategies for using polymer nanoparticles in the diagnosis and effective treatment of skin cancer, depending on the carrier molecule and the type of active substance being transported. It is possible to transport siRNA to inhibit the expression of key genes for melanoma cell proliferation [160]. Very promising results were obtained by Scopel et al. [161], who synthesized hybrid lipid-polymer nanoparticles with high affinity for the vitamin D3 receptor on the surface of B16 melanoma cells. Cell uptake experiments found the nanoparticles to effectively target B16 melanoma cells, thus offering a promising vehicle for delivering therapeutic agents for the treatment of melanoma. This method is therefore an excellent starting point for the development of targeted melanoma treatment protocols and the specific delivery of encapsulated therapeutic agents to other cells containing nuclear vitamin D receptors.

Wang et al. [162] synthesized polymeric nanoparticles that can carry protoporphyrin IX (PpIX), an effective photosensitizer that can selectively kill cancer cells following the activation of a special light source. Most importantly, cell viability studies revealed that PpIX-loaded polymersomes demonstrated low toxicity to healthy fibroblasts. However, nanocarriers and PDT exhibited considerable potential to selectively kill melanoma cells.

The use of a PDT photosensitive agent in combination with polymer nanocarriers was also proposed by Gamal-Eldeen et al. [163], who encapsulated indocyanine green (ICG) in polymer nanoparticles. Skin squamous cell carcinoma was induced in CD1 mice. The results clearly indicate that the ICG polymeric nanoparticles had high anticancer potential, with treatment resulting in decreased activity of TNF-L, COX-2 cyclooxygenase, and 5-LOX 5-lipoxygenase, which are involved in angiogenesis; treatment also resulted in enhanced apoptosis, caspase production, and histone acetylation.

Xia et al. [164] provided extremely valuable data on the use of the oncolytic peptide LTX-315 with polymeric nanocarriers in skin cancers. The results indicate that cRGD-functionalized polymersome chimeric (cRGD-CP) acts as a robust systemic delivery vehicle for LTX-315 that enhances the immunotherapy of B16F10 malignant melanoma in mice when combined with a CpG adjuvant and anti-PD-1. A significant decrease in the proliferation of cancer cells was observed, as well as a strong immune response, which was confirmed by increased secretion of, inter alia, IL-6, IFN-γ, and TNF-α. The obtained results can undoubtedly open a new way to the development of oncolytic peptides, which enables permanent cancer immunotherapy through systemic administration. 

#### 5.3.2. Dendrimers

When discussing polymeric nanocarriers in the context of their use in the treatment of skin cancer, dendrimers cannot be omitted. The latest research undoubtedly reveals the very high attractiveness of these molecules and the possibility of their wide application in the diagnosis and treatment of skin cancer. In their study, Ybarra et al. [165] proposed the use of dendrimers as nanocarriers for Vismodegib (VDG), an anticancer, first-in-class inhibitor of the Hedgehog signaling pathway, indicated for the treatment of locally advanced or metastatic basal cell carcinoma. The authors point to the development of interesting nanosystems with potential utility in local treatment of basal cell carcinoma. Xia et al. [166] used a combined strategy of chemotherapy and immunotherapy against murine B16F10 melanoma cells by encapsulating doxorubicin in the G4 PAMAM dendrimer with additionally integrated molecule cytosine–phosphate–guanine-based oligonucleotides followed by heparin coating. The compound showed improved treatment efficacy in primary melanoma tumor and lung metastases. Immune activation and multiple anti-metastatic effects of nanoparticles establishes a new therapeutic strategy for melanoma.

In Table 1, we have summarized all nanoparticles described in this review, taking into account the type of nanoparticles and study models.

## 6. Conclusions

The strategies given above illustrate just some applications of nanotechnology in the fight against skin cancer. The topic is extremely extensive and new molecules with good anticancer properties are constantly being developed, resulting in chemotherapeutic agents with improved effectiveness, bioavailability, and selectivity of action. The use of nanoparticles in oncology can overcome the disadvantages of standard therapy, allowing early detection of neoplastic changes and more accurate monitoring of treatment effectiveness. Indeed, the careful selection of appropriate nanocarriers and chemotherapeutic drugs has yielded clear dose reductions compared to standard therapy.

The International Agency for Research estimates that in twenty years, the number of skin cancer cases will grow to about 30 million, due to deteriorating environmental factors, demographic changes, and migration. Nanotechnology can play a key role in improving cancer treatment through the use of liposomes, metal nanoparticles, polymer nanoparticles, nanofibers, and carbon nanotubes, particularly for skin cancer, and can be used in the diagnosis of the early stages of melanoma and the treatment of skin cancer.

## Figures and Tables

**Figure 1 ijms-25-02165-f001:**
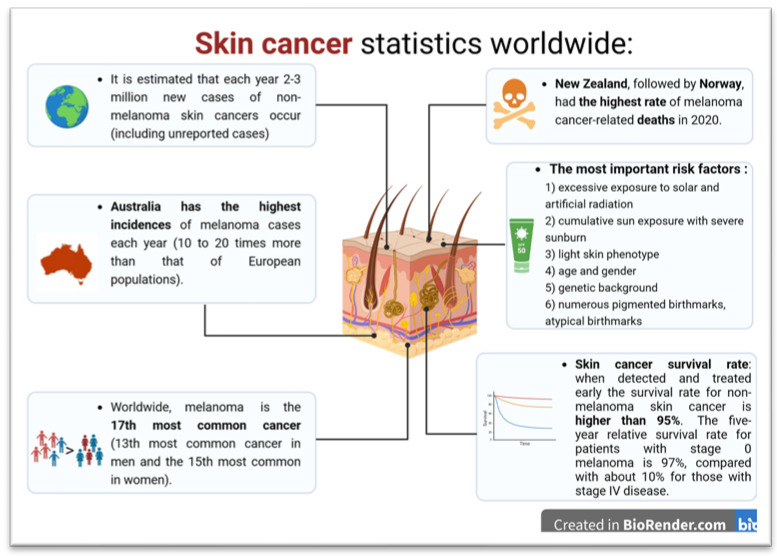
Skin cancer statistic based on WCRFI, WHO and American Cancer Society (The scheme was prepared based on [4,5,6,7]).

**Figure 2 ijms-25-02165-f002:**
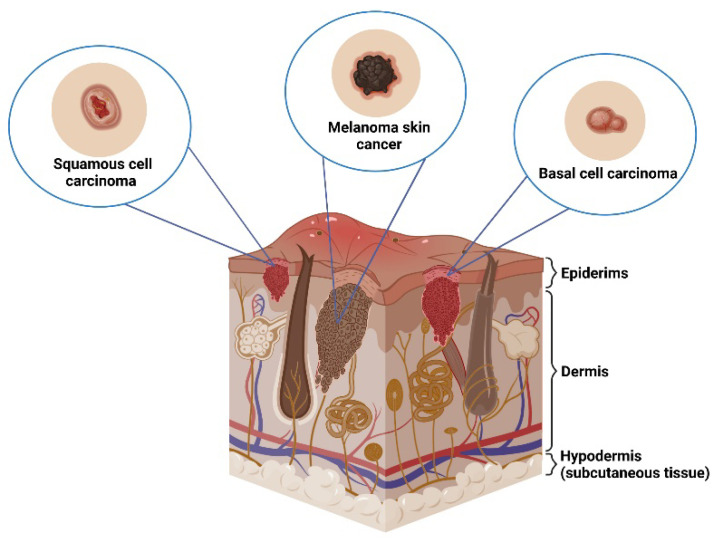
Types of skin cancers.

**Figure 3 ijms-25-02165-f003:**
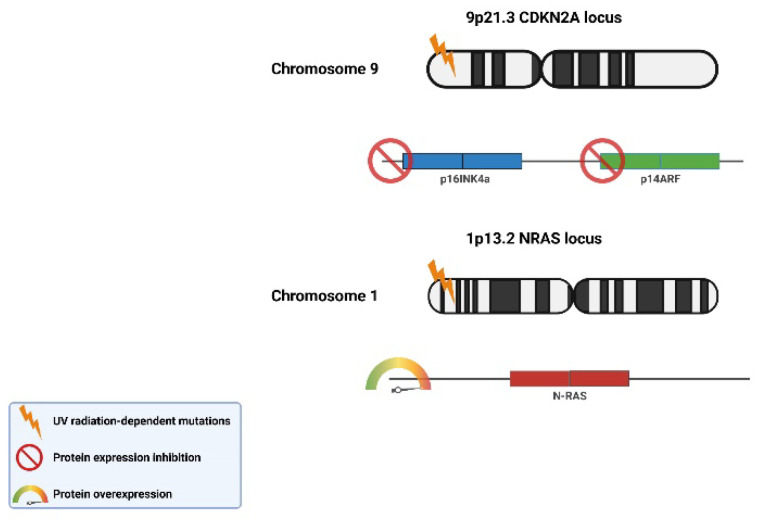
UV-dependent radiation mutations of tumor suppressor genes p16INK4a and p14ARF and proto-oncogene N-RAS characteristic of skin cancer. (The scheme was prepared based on [53]).

**Figure 5 ijms-25-02165-f005:**
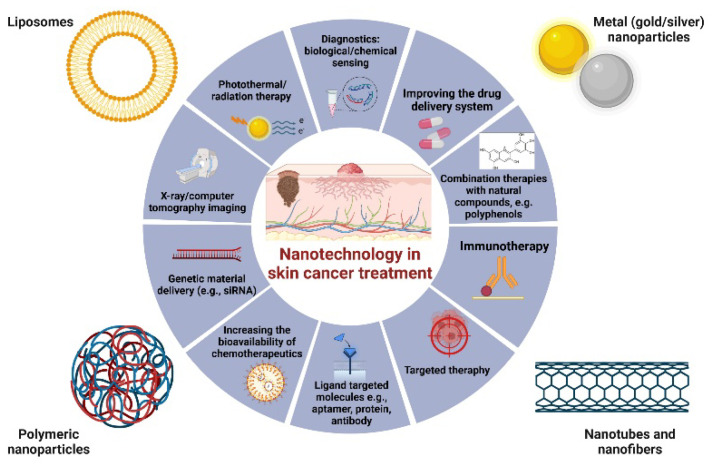
Strategies for the use of nanotechnology in the diagnosis and treatment of skin cancer. Examples of the use of liposomes, metal nanoparticles, and polymer nanoparticles, as well as nanotubes and nanofibrins, indicate the wide spectrum of biological activity of these compounds. The use of nanotechnology clearly increases the efficiency of classical chemotherapy by inter alia improving chemotherapeutic delivery systems, targeted therapies or combination therapies [70,71,72,73].

**Table 1 ijms-25-02165-t001:** Summary of nanoparticles described in the manuscript.

Type	Therapeutic Agent	In Vitro Cytotoxicity Study	In Vivo Animal Model	References
Lipid-based nanoparticles				
EGFR-targeted liposomes	5-FU	A431 and B16F10 cell lines	Immunosuppressed Swiss nude mice	[85]
Liposomes	Doxorubicin and celecoxib	A431 cell line	-	[86]
Aptamer liposomes	5-FU	TE 354.T cell line	-	[87]
Liposomes	Epigallocatechin gallatein	HDFa and HaCat cell lines	-	[88]
Liposomes	Quercetin and resveratrol	HDFa cell line	-	[89]
Cationic liposomes	Curcumin and STAT3 siRNA	A431, B16F10 cell lines	-	[91,92]
Liposomes	5-FU	B16-F10 cell line	-	[77]
Cubosomes	Paclitaxel	A431 cell line	Mice (female Balb/c nu/nu)	[166]
Liposomes	Doxorubicin	B16F10-OVA cell line	16F10 tumor- bearing mouse model	[95]
Lipid–polymer nanoparticles	Salinomycin	WM266-4 and A375 cell lines	Immunodeficient (SCID) mice	[96]
Liposomes	chlorin e61, 1’-dioctadecyl-3,3,3’,3’-tetramethylindotricarbocyanine iodide (DiR)	4T1 cell line	Balb/c mice	[99]
Liposomes	Methylene Blue and Acridine Orange	MET1 cell line	-	[100]
Liposomes	Human interferon b (HuIFNb) gene (IAB-1)	-	Stage IV or III melanoma patients	[108]
Liposomes	Lipovaxin-MM	-	Patient cohorts	[103]
Ethosomes	Doxorubicin and curcumin	B16 cell line	SD rats and C57BL/6	[107]
Ethosomes	Berberine chloride and evodiamine	B16 cell line	-	[108]
Ethosomes	Brucine	A375 cell line	-	[109]
Ethosomes	Nobiletin	A431 cells	Male Balb/C mice	[110]
Ethosomes	(−)-Epigallocatechin-3-gallate	A431 cells	Male Balb/C nude mice	[111]
Ethosomes	5-FU	SKMEL-2 cell line	Male Sprague Dawley rats	[112]
Ethosomes	Mitoxantrone	B16 cell line	Balb/C nude mice	[113]
Ethosomes	Metformin	-	Swiss albino mice	[114]
Ethosomes	Fisetin	-	Mice	[115]
Solid lipid nanoparticles	Docetaxel	SK-BR3, CT26 and 4T1 cell lines	Male C57BL/6 mouse and Sprague-Dawley rats	[117]
Solid lipid nanoparticles	5-FU	-	Male balb/C mice	[118]
Solid lipid nanoparticles	5-FU	B16F10 and A431 cell lines	-	[119]
Solid lipid nanoparticles	Dacarbazine	-	Wistar rats	[120]
Inorganic nanoparticles–metal nanoparticles				
Gold nanoparticles	Doxorubicine	A549 and B16F10 cell lines	C57BL6/J mice	[123]
Gold nanoparticles	*Zinnia elegans* plant extract	SK-OV-3, A549, and MCF-7 cell lines	C57BL6/J mice	[125]
Gold nanoparticles	Shikimoyl ligand	B16F10 cell line	C57BL6/J mice	[126]
Gold nanoparticles	5-FU	A431 cell line	C57BL6/J mice	[129]
Silver nanoparticles	5-FU	A431 cell line	C57BL6/J mice	[130]
Gold nanoparticles	Doxorubicin and nisin	-	BALB/c mice	[131]
Gold nanoparticles	Methotrexate	Human skin equivalents (HSEs)	-	[132]
Gold nanoparticles	*Vitis vinifera* peel polyphenols	A431 cell line	-	[134]
Gold nanoparticles	*Siberian ginseng*	B16 cell line	-	[135]
Silver nanoparticles	*Trapa natans* extract	A431 cell line	-	[136]
Gold nanocages	Monophosphoryl lipid and indocyanine green	B16-F10 cell line	C57BL/6 mice	[138]
Silver based nanohybrids	Zinc phthalocyanine tetrasulfonate (ZnPcS4) and folic acid	A375 cell line	-	[139]
Gold nanocages	anti-MUC18 single-chain antibod	A375 cell line	-	[140]
Gold nanocages/SiO_2_	Aptamer	Mcf-7 and NIH 3T3 cell lines	-	[141]
Nanotubes and nanofibers				
Carbon nanotubes	Camptothecin	MDA-MB-231 cell line	-	[146]
Chitosan/dodecyl sulfate nanofibers	Pyrazoline H3TM04	B16-F10 cell line	-	[148]
Nanofibers	AuNPs and curcumin	3 T3 and A431 cell lines	-	[149]
Nanofibers	5-FU	L929 and B16-F10 cell lines	-	[151]
Nanofibers	Etoposide and Methotrexate	SKMEL-3 cell line	-	[152]
Polymeric nanoparticles				
Polymersomes	Doxorubicin	B16 cell line	C57BL/6	[158]
Lipid–polymer hybrid nanoparticles	Vitamin D3	B16 cell line	-	[161]
Polymersomes	Protoporphyrin IX-	A375 cell line	-	[162]
Polymeric nanoparticles	Indocyanine green	-	CD1 mice	[163]
Polymersomes	Oncolytic peptide LTX-315	B16-F10 cell line	B16F10 tumor-bearing mice	[164]
PAMAM-dendrimers	Vismodegib	HaCaT cell line	-	[165]
G4 PAMAM-dendrimers	Doxorubicin	B16-F10 cell line	B16F10 tumor-bearing mice	[166]

## Data Availability

No new data were created.

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
