# Peer review of "Nanotechnology as a Promising Method in the Treatment of Skin Cancer"

_ijms, 2024, doi:10.3390/ijms25042165_

Round 1

Reviewer 1 Report

Comments and Suggestions for Authors

The authors describe a review article on the use of nanocarriers for treatment of skin cancer. The topic is overall important, since new strategies are always being sought for treatment of this disease. However, I have some comments to be addressed:

1- Extensive language editing is required.

2- Please replace the name" melanoma malignant" with "melanoma" throughout the manuscript, since melanoma is malignant by default, and no one refers to melanoma as "malignant melanoma" or "melanoma malignant"

3- In the titles of figures 3 and 4, the authors placed references as [53], [54], [62] etc..., please state whether permissions for reuse of these figures have been obtained.

4- I am not sure why the authors have only stated liposomal applications, without referring to the newer generations of liposomes such as penetration enhancer vesicles and ethosomes in treatment of skin cancer. Some examples are doi: 10.1016/j.ejps.2019.104972, doi: 10.1038/s41598-021-99756-1, doi: 10.3109/10837450.2016.1146294

5- Title 11 carbon nanotubes and nanofibers, please separate them into two different sections, as they have completely different properties, and I don't see how they are relevant to each other to be placed in one section

6- Please remove the reference from the conclusion section, and re-write it to highlight the carriers described in the review article. Also, include future perspective in the conclusion section

4- 

Comments on the Quality of English Language

Extensive editing is required, since the manuscript is full of grammatical and spelling mistakes.

Author Response

The response to the Reviewer Comments

Comment 1.- Extensive language editing is required

Response 1 We thank the Reviewer for his comments on the manuscript. We have made corrections to the publication in accordance with your suggestions. The English language has been checked and corrected by a professional translator and native speaker of English. We attach a certificate issued by the translator.

Comment 2 Please replace the name" melanoma malignant" with "melanoma" throughout the manuscript, since melanoma is malignant by default, and no one refers to melanoma as "malignant melanoma" or "melanoma malignant"

Response 2  Thank you very much for this valuable comment. In the manuscript, the name malignant melanoma was replaced by the term melanoma

Comment 3 In the titles of figures 3 and 4, the authors placed references as [53], [54], [62] etc..., please state whether permissions for reuse of these figures have been obtained.

Response 3 All schemes and figures in the manuscript are original and they were prepared by authors. We placed references below the figures because the schemes were prepared based on the literature.

Comment 4: I am not sure why the authors have only stated liposomal applications, without referring to the newer generations of liposomes such as penetration enhancer vesicles and ethosomes in treatment of skin cancer. Some examples are doi: 10.1016/j.ejps.2019.104972, doi: 10.1038/s41598-021-99756-1, doi: 10.3109/10837450.2016.1146294

Response 4: Thank you very much for the comment. In fact, we completely omitted ethosomes from our review paper. This error has been rectified as suggested and a subsection on the use of ethosomes in the treatment of skin cancer has been added (section no. 5.1.2). In addition, we wanted to point out that within the chapter on lipid-based nanoparticles (section no. 5.1), we have added a short subsection on solid lipid nanoparticles (section no. 5.1.3).

Comment 5: Title 11 carbon nanotubes and nanofibers, please separate them into two different sections, as they have completely different properties, and I don't see how they are relevant to each other to be placed in one section.

Response 5: As suggested by the reviewer, we split into two sections between carbon nanotubes and nanofibers. We have included the two sections in the chapter on Inorganic nanoparticles (section no. 5.2).

Comment 6 Please remove the reference from the conclusion section, and re-write it to highlight the carriers described in the review article. Also, include future perspective in the conclusion section

Response 6: We have removed the reference from the conclusion section and corrected the conclusion.

Thank you for your suggestions and comments.

Reviewer 2 Report

Comments and Suggestions for Authors

I have reviewed the review article “Nanotechnology as a promising method in the treatment of skin cancer”. I have the following suggestions:

1.      The language needs to be improved. The wording needs to be more fluent, and the authors should correct the typographic errors.

2.      It’s a medical article which appears to be written by non-medical authors. The use of medical terminologies is not appropriate in many areas (e.g. melanoma malignant, instead of malignant melanoma or simply melanoma). The manuscript can be improved by a review/editing by an experienced physician (preferably a pathologist/dermatopathologist or dermato-oncologist or dermatologist). There are a few statements which are not accurate and need to be corrected. Example: “Chemotherapy is, slightly more often used in the treatment of some rarer types of skin cancers, such as basal cell carcinoma or squamous cell carcinoma, especially in advanced cases”. Note: BCC and SCC are not rare.

3.      A summary table in the end can improve the manuscript.

4.      The first 4 sections are too long, as they have little to do with the main focus of the review, i.e. nanotechnology. These sections discuss the general epidemiology and pathophysiology of cutaneous malignancies, which are available in many previous reviews. These 4 parts need to be shortened with editing focused on removing repetition of information.

5.      Section 5 onwards also need to be shortened. There is lot of repetition, and the sections are not well-organized to deliver the desired information.

Thank you!

Comments on the Quality of English Language

Moderate editing of language required.

Author Response

The response to the Reviewer 2 Comments

Comment 1: The language needs to be improved. The wording needs to be more fluent, and the authors should correct the typographic errors.

Response 1: The authors thank the Reviewer for his valuable comments on the manuscript. We have made corrections to the publication in accordance with your suggestions. The English language has been checked and corrected by a professional translator and native speaker of English. We have corrected errors in the article. We attach a certificate issued by the translator.

Comment 2: It’s a medical article which appears to be written by non-medical authors. The use of medical terminologies is not appropriate in many areas (e.g. melanoma malignant, instead of malignant melanoma or simply melanoma). The manuscript can be improved by a review/editing by an experienced physician (preferably a pathologist/dermatopathologist or dermato-oncologist or dermatologist). There are a few statements which are not accurate and need to be corrected. Example: “Chemotherapy is, slightly more often used in the treatment of some rarer types of skin cancers, such as basal cell carcinoma or squamous cell carcinoma, especially in advanced cases”. Note: BCC and SCC are not rare.

Response 2: Thank you for your comment. We agree that the terminologies should be more medical, so we have corrected many errors, especially the melanoma terminology. We replaced malignant melanoma with melanoma.

Comment 3: A summary table in the end can improve the manuscript.

Response 3: Following the reviewer's suggestion, we have included a table summarizing all the nanoparticles described, detailing the type of study (in vitro/in vivo) and the research model.

Comment 4: The first 4 sections are too long, as they have little to do with the main focus of the review, i.e. nanotechnology. These sections discuss the general epidemiology and pathophysiology of cutaneous malignancies, which are available in many previous reviews. These 4 parts need to be shortened with editing focused on removing repetition of information.

Response 4: Following the reviewer's suggestion, the authors shortened and condensed the chapters, especially on skin cancer, epidemiology and risk factors. We removed a lot of repetition and focused on the nanotechnology.

Comment 5: Section 5 onwards also need to be shortened. There is lot of repetition, and the sections are not well-organized to deliver the desired information.

Response 5: Thank you for this valuable attention. We have decided to revise section 5 as suggested. Some paragraphs have been removed due to unnecessary repetitions. Moreover, we decided to edit the entire section of section 5. We introduced main chapters on the type of nanoparticles: 5.1. Lipid-based nanoparticles, 5.2. Inorganic nanoparticles and 5.3. Polymer-based nanoparticles. Each of the above-mentioned chapters is divided into appropriate subchapters describing nanoparticles belonging to the specific types described. We believe that the new division of this section will improve the clarity of the article and improve its reception for the reader.

Thank you for your suggestions and comments.

Round 2

Reviewer 2 Report

Comments and Suggestions for Authors

The authors have made many important corrections and edits. However, there is still room for improvement. There are still sentences like "melanoma is one of the most malignant cancer", terms which are usually not used that way. The authors may want to say "melanoma is one of the most aggressive cutaneous malignancy". I think the manuscript can be improved further by review from a dermatopathologist or dermatologist, so the medical terms can be corrected. Another round of corrections and summarization is needed.

Thank you.

Comments on the Quality of English Language

Minor corrections are required.

Author Response

Thank you very much for the second round of reviews. Your comments are valuable to us, so we have improved the manuscript. As the authors of a review on nanotechnology in the treatment of skin cancer, we wanted to draw attention to new therapeutic possibilities. The aim of the work was to write a non-medical article, understandable to all scientists. We have corrected the indicated sentences in the manuscript and modified chapter 2. Skin cancer and chapter 3. Photoaging as a risk factor for the development of skin cancer. The references in the manuscript have been corrected. We have corrected english language in the article.All changes can be tracked in track changes mode. In addition, we have marked the corrected parts of the manuscript in blue. We hope that the corrections introduced will be satisfactory and the manuscript will be accepted.

Thank you!
